# Upregulation of Wheat Heat Shock Transcription Factor *TaHsfC3-4* by ABA Contributes to Drought Tolerance

**DOI:** 10.3390/ijms25020977

**Published:** 2024-01-12

**Authors:** Zhenyu Ma, Baihui Zhao, Huaning Zhang, Shuonan Duan, Zihui Liu, Xiulin Guo, Xiangzhao Meng, Guoliang Li

**Affiliations:** 1Institute of Biotechnology and Food Science, Hebei Academy of Agriculture and Forestry Sciences/Hebei Key Laboratory of Plant Genetic Engineering, Shijiazhuang 050051, China; mazhenyuqqtt@163.com (Z.M.); zbaihui0306@163.com (B.Z.); 13821201341@163.com (H.Z.); duanshuonan@haafs.org (S.D.); liuzihui1978@163.com (Z.L.); myhf2002@163.com (X.G.); 2College of Life Sciences, Hebei Normal University, Shijiazhuang 050024, China

**Keywords:** wheat, drought, abscisic acid, heat shock transcription factors, *TaHsfC3-4*

## Abstract

Drought stress can seriously affect the yield and quality of wheat (*Triticum aestivum*). So far, although few wheat heat shock transcription factors (Hsfs) have been found to be involved in the stress response, the biological functions of them, especially the members of the HsfC (heat shock transcription factor C) subclass, remain largely unknown. Here, we identified a class C encoding gene, *TaHsfC3-4*, based on our previous omics data and analyzed its biological function in transgenic plants. *TaHsfC3-4* encodes a protein containing 274 amino acids and shows the basic characteristics of the HsfC class. Gene expression profiles revealed that *TaHsfC3-4* was constitutively expressed in many tissues of wheat and was induced during seed maturation. *TaHsfC3-4* could be upregulated by PEG and abscisic acid (ABA), suggesting that this Hsf may be involved in the regulation pathway depending on ABA in drought resistance. Further results represented that TaHsfC3-4 was localized in the nucleus but had no transcriptional activation activity. Notably, overexpression of *TaHsfC3-4* in *Arabidopsis thaliana pyr1pyl1pyl2pyl4* (*pyr1pyl124*) quadruple mutant plants complemented the ABA-hyposensitive phenotypes of the quadruple mutant including cotyledon greening, root elongation, seedling growth, and increased tolerance to drought, indicating positive roles of TaHsfC3-4 in the ABA signaling pathway and drought tolerance. Furthermore, we identified TaHsfA2-11 as a TaHsfC3-4-interacting protein by yeast two-hybrid (Y2H) screening. The experimental data show that TaHsfC3-4 can indeed interact with TaHsfA2-11 in vitro and in vivo. Moreover, transgenic *Arabidopsis TaHsfA2-11* overexpression lines exhibited enhanced drought tolerance, too. In summary, our study confirmed the role of TaHsfC3-4 in response to drought stress and provided a target locus for marker-assisted selection breeding to improve drought tolerance in wheat.

## 1. Introduction

With the continuous changes in the global climate, droughts caused by water shortages severely restrict crop yields and quality, posing an increasing threat to food security worldwide [1,2]. In the past decade, an average loss of 13.7% in global cereal production was caused by water scarcity [3]. Upon exposure to drought, plants show a series of impairments, including cell injury caused by the generation of reactive oxygen species (ROS) and the increase in intracellular temperature, which eventually leads to an increase in the viscosity of cellular contents, protein aggregation, and denaturation [4]. At the same time, dehydration causes cell shrinkage followed by a significant decrease in cell volume and a large accumulation of solutes, resulting in toxicity that negatively affects the function of certain enzymes, often leading to the reduction of water use efficiency (WUE) and the inhibition of photosynthesis [4]. In addition, plants will also have obvious morphological changes under prolonged dehydration, exhibiting leaf coiling accompanied by wilting and bleaching [5]. Usually, drought occurs when the soil moisture cannot meet the needs of plant growth and development, so areas with uneven rainfall distribution and groundwater shortages are most vulnerable to drought [6]. Wheat is an important cereal crop planted in semi-arid and arid areas, and its growth and development are often subjected to drought stress, resulting in reduced production [7]. However, unlike model plants, much less is known about the genetic basis and molecular mechanism of drought resistance in wheat [8]. Therefore, it is very necessary to study the molecular mechanism of wheat in response to drought stress and to explore the excellent drought resistance genes to ensure sustainable food production and global food security.

The response of crops to drought stress is sophisticated and multi-dimensional, including “drought escape”, “drought avoidance”, and “drought tolerance” [9,10,11]. In terms of improving drought resistance, plants mainly cope with drought stress through a series of molecular, cellular, and physiological regulations. For example, the induction/repression of various genes during drought stress could cause the accumulation of osmotic substances, improvement of antioxidant systems, reduction in transpiration, inhibition of shoot growth, and decrease in tillering [4]. In addition, drought-induced ABA signaling mediated the regulation of stomatal movement, the modulation of the root system, and the expression of various stress-related genes [12]. Over the past two decades, great progress has been made in the study of ABA signaling in improving the drought resistance of crops [1,2,13,14]. Under well-watered conditions, PP2C-As (clade A protein phosphatase 2 proteins) interact with group III SnRK2s (sucrose non-fermenting 1-related protein kinase 2 proteins) and inhibit ABA signaling [15]. In response to drought stress, ABA accumulates rapidly and binds to PYL receptors PYR1 (pyrabactin resistance 1), PYL (PYR1-like), or RCAR (regulatory components of ABA receptor) [16]. The ABA-bound PYL receptors competitively bind to PP2C-As (clade A protein phosphatases type 2C) to form ternary complexes [17,18]. PP2C-As inhibition in turn releases SnRK2s, which phosphorylate downstream AREBs/ABFs (ABA-responsive element-binding protein/ABA-responsive element-binding factors) to initiate protective responses [19]. In fact, ABA also has important roles in regulating CDPK (calcium-dependent protein kinases) and MAPK (mitogen activated protein kinases) signaling pathways [20,21]. Taken together, ABA is of prime importance due to its involvement in the regulation of many downstream drought-responsive genes, thereby enhancing plant adaptability to water deficit conditions.

In recent years, many key genes and transcriptional regulators have been discovered in drought tolerance research [1,2]. Among these regulatory genes, transcription factors (TFs) are particularly promising targets for the genetic improvement of crops in resistance to environmental stress because the engineering of individual TFs can affect various related genes in their respective regulons, ultimately influencing the physiological process or phenotypic traits for which they are responsible [4]. Compared with those in the genomes of animals and yeasts, genes encoding TFs may occupy a large proportion of the plant genome, and many TFs generally belong to large gene families, such as the Hsf family, which is classified into three major classes, namely, HsfA, HsfB, and HsfC [22,23,24,25]. Typically, the proteins encoded by plant Hsfs have relatively conserved structural characteristics, including a typical N-terminal DBD (DNA-binding domain) characterized by a central HTH (helix-turn-helix) motif, an OD (oligomerization domain) with an HR-A/B region which possesses a bipartite heptad pattern of hydrophobic amino acid residues, and a NLS (nuclear localization signal) for regulating the assembly of the nuclear import complex and the C-terminal activation domain formed of AHA (aromatic, hydrophobic, acidic) amino acid residues [24,26,27]. Although Hsfs in plants share a well-conserved modular structure, their remarkable diversification between monocots and dicots reflects their numerous functions in the complex signaling pathways and response networks [28]. For instance, HsfB1 functions as a coactivator, cooperating with class A Hsfs and HAC1/CBP (CREB-binding protein) during ongoing heat stress in tomato plants, while *Arabidopsis* HsfB1 was found to act as a transcriptional repressor and negatively regulate the expression of heat-inducible Hsfs and Hsps (heat shock proteins) [29,30]. Interestingly, Hsfs of the C2 subclass appear to be monocot-specific, whereas subclasses A9, B3, and B5 are absent in monocot species [27]. As the indispensable components of the signal transduction pathway, plant Hsfs mediate stress-related gene expression in response to various abiotic stresses by binding to the heat shock element (HSE) located in the promoter regions of these associated genes [31]. However, because bread wheat (*Triticum aestivum*, AABBDD) is a hexaploid wheat with a more complex Hsf family structure than that of other species, only a few members of the wheat Hsf family have been functionally analyzed [32]. In view of the great potential of Hsfs in improving wheat’s tolerance against environmental stresses, the identification of new Hsfs will accelerate conventional crop breeding for improved stress resistance.

In recent years, functional genomics approaches, such as transcriptomics, have become important tools for understanding TFs in responses to various abiotic stresses, including drought stress [4]. Our previous studies have demonstrated that there is a new subclass in wheat with 13 TaHsfC3 members which did not have any orthologous genes in *Arabidopsis*, rice, and maize, and the roles of this subclass are currently unknown. Of particular note, the encoding gene *TaHsfC3-4* was highly induced by drought treatment and ABA compared with other 12 TaHsfC3s in wheat leaves [32]. In this study, we further investigated whether TaHsfC3-4 mediates ABA signaling and drought tolerance and aim to discover valuable genetic targets for breeding drought-tolerant wheat cultivars. 

## 2. Results

### 2.1. Characteristics and Bioinformatics Analysis of TaHsfC3-4

To further understand the biological function of *TaHsfC3-4*, the complete coding sequence of the gene was obtained from the leaves of wheat variety C6005 (GenBank: OQ680124.1). This gene encodes 274 amino acids and is predicted to have a protein molecular weight of 30 kDa. The TaHsfC3-4 protein has the basic characteristics similar to the HsfC from other species, including DBD, OD and NLS, and also lacks AHA (Appendix A). The three-dimensional structure of TaHsfC3-4 constructed by the online tool swiss-model software (https://swissmodel.expasy.org/, accessed on 3 January 2024) also indicates that this gene has 88.24% similarity to the TaHsfC2-like model template and belongs to the HsfC family (Figure 1A). Next, the phylogenetic tree was constructed using TaHsfC3-4 and HsfCs from other species. The results indicated that TaHsfC3-4 was highly homologous to other TaHsfC3 members in wheat and was a typical HsfC gene (Appendix A).

As the connecting points between upstream regulatory factors and downstream response genes, *cis*-elements located in promoters can reflect the biological function of associated genes [33]. Therefore, the promoter sequence with 2855 bp length, upstream of the ATG codon of *TaHsfC3-4*, was isolated from wheat genomic DNA and predicted by using the PlantCARE database [34]. The results showed that there were twelve ABA-responsive cis-elements, one salicylic acid-responsive cis-element, one auxin-responsive cis-element, two gibberellin-responsive cis-elements, six MeJA-responsive cis-elements, two MYB-binding sites involved in drought inducibility, and one cis-acting regulatory element related to seed-specific regulation (Figure 1B). The greatest number of ABA-responsive *cis*-elements was found in the promoter of *TaHsfC3-4*, suggesting that ABA may induce or activate *TaHsfC3-4* expression. 

### 2.2. Expression Patterns of TaHsfC3-4

To explore the expression patterns of *TaHsfC3-4*, RT-qPCR (real-time quantitative PCR) was used to determine the transcript levels of *TaHsfC3-4* in different wheat tissues at seedling and reproductive stages. The highest expression of *TaHsfC3-4* was found in seeds, followed by that in young leaf, mature root, young root, young shoot, stamen, mature leaf, mature shoot, and pistil (Figure 2A). For wheat and other gramineous crops, their reproductive stages, such as flowering and seed development, are especially sensitive to drought stress [35,36,37,38]. Therefore, we further examined the expression of *TaHsfC3-4* during seed maturation. The RT-qPCR results showed that *TaHsfC3-4* was substantially induced and reached the peak at 27 days after anthesis (Figure 2B).

Next, in order to confirm the induction of *TaHsfC3-4* in response to ABA and dehydration treatments, we detected the expression levels of *TaHsfC3-4* in young seedlings of wheat which were subjected to exogenous ABA and PEG6000, respectively. The results of the RT-qPCR analyses showed that the transcript levels of *TaHsfC3-4* increased and peaked at 12 h in both the roots and leaves under exogenous ABA treatment (Figure 2C,D). Moreover, it was found that *TaHsfC3-4* was also induced by dehydration treatment, and the peak time of *TaHsfC3-4* in roots and leaves was 12 h and 8 h, respectively (Figure 2E,F). Together, ABA or dehydration treatment strongly promoted the expression of *TaHsfC3-4*.

### 2.3. TaHsfC3-4 Encodes a Nucleus-Localized Protein with No Transcriptional Activation Activity

As shown in Appendix A, TaHsfC3-4, like other HsfCs, has NLS and also lacks the AHA motif in terms of the protein structure [39], so the true transcriptional activity and localization of its protein is worth considering. Firstly, by using the GAL4-based yeast system, we investigated the transcriptional activity of TaHsfC3-4. As expected, all the transformants grew well on the yeast synthetic drop-out medium lacking Trp (SD/-Trp). However, only the yeast transformed with pGBKT7-p53 grew well on the yeast synthetic drop-out medium lacking Trp, His, and Ade (SD/-Trp/-His/-Ade) and could catalyze X-α-Gal (5-bromo-4-chloro-3-indolyl-α-d-galactopyranoside acid); the yeast harboring pGBKT7-TaHsfC3-4 or pGBKT7 barely grew at all on SD/-Trp/-His/-Ade medium (Figure 3A). Next, the *GFP* sequence was fused to the *TaHsfC3-4* and then transiently expressed in the leaf epidermal cells of *N. benthamiana* to detect the subcellular localization of the fusion protein. It could be seen that the fluorescence signal of TaHsfC3-4-GFP was obviously concentrated in the nucleus, whereas the fluorescence signal of free GFP was distributed throughout the whole cell. The results indicated that TaHsfC3-4 may function in the nucleus (Figure 3B). 

### 2.4. TaHsfC3-4 Positively Regulates ABA Sensitivity during Post-Germination Growth

To further determine the role of *TaHsfC3-4* in the ABA signaling pathway, we generated three transgenic plants overexpressing *TaHsfC3-4* in the *pyr1pyl124* quadruple mutant (*139-18*, *125-10,* and *23-8*), which has mutations in multiple PYL genes and shows strong ABA-resistant phenotypes (Appendix A). Furthermore, we observed the ABA-related phenotype of the transgenic lines, the *pyr1pyl124* quadruple mutant, and *Col-0* during post-germination seedling development. As shown in Figure 4A,C, in terms of cotyledon phenotypes and the percentages of green cotyledons, there was no significant difference among the transgenic lines, the *pyr1pyl124* quadruple mutant, and *Col-0* under 0 μM ABA treatment. However, as previously reported, the *pyr1pyl124* quadruple mutants were less sensitive to the same ABA treatment than *Col-0* when grown on MS plates with 0.5 μM ABA [15]. Notably, overexpression of *TaHsfC3-4* in *pyr1pyl124* complemented the ABA-hyposensitive phenotypes of the quadruple mutant and showed the same or lower percentages of green cotyledons when compared with those of *Col-0* (Figure 4B,D). Correspondingly, the phenotype of root elongation was also assayed and compared at the stage of post-germination seedling development. It was found that no obvious difference in terms of root elongation was observed among the transgenic lines, *pyr1pyl124* quadruple mutant, and *Col-0* under 0 μM ABA treatment (Figure 5A,C). However, the transgenic lines were more sensitive to ABA treatment like *Col-0* when compared with *pyr1pyl124* quadruple mutant under ABA treatment. Together, these data suggested that the overexpression of *TaHsfC3-4* in *pyr1pyl124* could have complemented the post-germination ABA sensitivity of the quadruple mutant (Figure 5B,D). 

### 2.5. The Overexpression of TaHsfC3-4 Enhanced the Tolerance of pyr1pyl124 Quadruple Mutant to Drought

Since ABA is an important phytohormone for plant drought tolerance, and since the above results demonstrated that TaHsfC3-4 plays a positive role in ABA signaling, we inferred that *TaHsfC3-4* overexpression may also confer drought tolerance in transgenic plants. To further test our hypothesis, identical numbers of transgenic plants, the *pyr1pyl124* quadruple mutant, and *Col-0* were grown in soil mix for three weeks before water was withheld. Except that the *pyr1pyl124* quadruple mutant shows delayed growth, as previously reported [15], there were no obvious differences in the growth performance between transgenic plants and *Col-0* plants under well-watered conditions (Figure 6A). After drought and rewatering, the transgenic plants recovered better than the *Col-0* plant, while the *pyr1pyl124* quadruple mutant had more withered and dead leaves than the *Col-0* plant (Figure 6B). Consistent with this observation, the content of chlorophyll in transgenic plants was also significantly higher than that in the *pyr1pyl124* quadruple mutant and the *Col-0* plant, indicating a lower degree of influence in transgenic lines after drought (Figure 6C). Meanwhile, we investigated the activities of peroxidase (POD), superoxide dismutase (SOD), and malondialdehyde (MDA) production in transgenic lines, the *pyr1pyl124* quadruple mutant, and the *Col-0* plant under drought treatment or not. The results showed that under well-watered conditions, there was no difference among the transgenic lines, the *pyr1pyl124* quadruple mutant, and the *Col-0* plant. Unexpectedly, there was also no significant difference in terms of the POD activity among the transgenic lines, *pyr1pyl124* quadruple mutant, and *Col-0* plant after drought treatment (Figure 6D). However, a higher activity of SOD and lower production of MDA in transgenic lines was detected when subjected to drought (Figure 6E,F).

### 2.6. TaHsfA2-11 Interacts with TaHsfC3-4 and Is Involved in Improving Drought Tolerance

To elucidate the regulatory mechanism of TaHsfC3-4 in the drought stress response, a homogenized cDNA library of wheat was used to screen for potential interacting proteins (Appendix A). Sequencing revealed that a member of the HsfA subclass named TaHsfA2-11 was identified as one of the potential TaHsfC3-4-interacting proteins. To confirm the interaction between TaHsfC3-4 and TaHsfA2-11, we obtained the full-length coding sequence (CDS) of *TaHsfA2-11* and fused it with the GAL4 activation domain (AD) to perform a Y2H assay. As shown in Figure 7A, TaHsfC3-4 is physically associated with TaHsfA2-11 in the Y2H system. In addition, to verify the interaction between TaHsfC3-4 and TaHsfA2-11 in vivo, split firefly luciferase complementation (LCI) assays were also performed by transiently co-expressing both proteins in the leaf epidermal cells of *N. benthamiana*. Strong interaction signals were detected in *N. benthamiana* leaves transiently co-expressing TaHsfC3-4-nLUC and cLUC-TaHsfA2-11 but not in those co-expressing TaHsfC3-4-nLUC and cLUC, cLUC -TaHsfA2-11 and nLUC, or nLUC and cLUC (Figure 7B). 

Given the function of TaHsfC3-4 in drought, we speculated that TaHsfA2-11 is also involved in the response to drought. To test this hypothesis, three transgenic plants overexpressing *TaHsfA2-11* in *Col-0* (*3-4*, *7-44*, and *8-49*) were generated, and the role of TaHsfA2-11 in water deficit conditions was analyzed. Experimental results indicated that all the transgenic plants exhibited a similar growth phenotype as *Col-0* under well-watered conditions. After drought stress, wildtype plants gradually wilted and their leaves became curled, while three transgenic lines remained turgid with fewer withered leaves (Figure 7C). These results indicate that TaHsfC3-4 is involved in improving the drought resistance of plants, probably through the heterologous interaction with TaHsfA2-11. Of course, the discovery of this molecular mechanism still requires more effort and time. 

## 3. Discussion

### 3.1. TaHsfC3-4 Is a Typical HsfC Gene

It is now well-established that Hsf plays a crucial role in protecting against stress damage by regulating the transcript levels of stress-responsive genes [28]. Compared with few Hsf members in model plants such as *Arabidopsis*, tomato plants, and rice, a large number of Hsfs have been isolated and identified at a genome-wide scale in wheat [27]. Based on the analysis of the reference genome, more HsfC genes were discerned in monocots, such as in wheat, and up to 5 and 7 genes were classified into subclasses C1 and C2, respectively. In fact, there are more genes assigned into subclasses C1 and C2 in wheat (Appendix A). Meanwhile, the presence of subclass C2 is considered to be an important feature that distinguishes monocot plants from eudicots [27]. With the development of comparative genomics, a new clade named HsfC3 with 13 members in wheat has been found, and the function of these genes urgently needs to be explored [32]. In the present study, we successfully cloned an HsfC3 gene from wheat based on our previous omics data and named it *TaHsfC3-4* here. Like other HsfBs and HsfCs, TaHsfC3-4 has the basic characteristics but also lacks the AHA domain in terms of its protein structure (Appendix A and Figure 1A) [39]. Indeed, TaHsfC3-4 is localized to the nucleus and has potential in the regulation of downstream stress response genes, while it does not have transcriptional activation activity in yeast cells (Figure 3A,B). It is worth mentioning that the characteristic tetrapeptide LFGV in the C-terminal domain of class HsfB (except HsfB5), functioning as repressor domain (RD), was also not been found in TaHsfC3-4 [40,41,42]. Therefore, we speculated that TaHsfC3-4 may act as a coactivator to co-regulate the expression of drought response genes with other regulatory factors. However, we cannot rule out that TaHsfC3-4 could exert a negative regulation of gene expression as an individual transcriptional suppressor through an unknown repressor domain.

### 3.2. TaHsfC3-4 Enhances Tolerance to Drought Depending on ABA Signaling Pathway

The phytohormone ABA plays a vital role in plant adaptation during droughts and other stresses [43]. ABA can trigger a series of physiological and biochemical processes such as stomatal closure, root system modulation, transcriptional activation of gene expression, post-translational modification of proteins, and metabolic alterations in order to cope with environmental stresses, particularly, drought [44]. Thus, many ABA-induced genes are also involved in plants’ response to drought. In our study, the greatest number of ABA-responsive *cis*-elements was found in a promoter of *TaHsfC3-4*, suggesting that ABA may induce or activate *TaHsfC3-4* expression (Figure 1B). Next, the gene expression profiles showed that the *TaHsfC3-4* transcripts were induced by ABA or dehydration treatment strongly in both the leaves and roots of wheat (Figure 2C–F), which is consistent with our previous transcriptomic data [32]. Similarly, in a recent study, three *TaHsfC2a* homoeologous genes were markedly upregulated by drought and ABA treatment [45]. Since TaHsfC3-4 shared higher similarity with HsfC2a, similar results were to be expected (Appendix A and Figure 1A). Additionally, ABA also regulates many important aspects in plants, including seed germination and dormancy, post-germination seedling growth, leaf senescence, and so on [44]. Our experimental results show that the transcript levels of *TaHsfC3-4* in seeds was highest among the tested tissues and substantially induced during seed maturation (Figure 2A,B), suggesting that *TaHsfC3-4* is involved in seed maturation. Therefore, in the future, some of our efforts will be devoted to investigating the potential molecular mechanisms of TaHsfC3-4 regulated by ABA during seed maturation.

The multiplicity of Hsfs in plants is predicted to be related to their numerous functions in the complex signaling pathway and response networks [28]. Although most plant Hsfs are regulated by heat stress, including up- and downregulation, there are several Hsfs that have been reported to be involved in the plant response to drought via ABA dependent/independent pathways. For example, *Arabidopsis* HSFA1b has been shown to be a key factor of drought/dehydration tolerance, and its function is independent on the expression of ABA- or dehydration-responsive genes [46]. In contrast, *HSFA3* is regulated by a heat- and drought-responsive TF *DREB2A* in the drought stress response [27]. In addition, Xiang et al. showed that rice *OsHsfB2b* was strongly induced in response to ABA and dehydration treatments and functioned as a negative regulator during drought stress [47]. After analyzing the ABA responses in seed germination, cotyledon greening, root growth, and drought tolerance in *HSFA6b*-null mutant, *HSFA6b*-OE, and dominant-negative effect (*HSFA6b*-RD) lines, Huang et al. found that *HSFA6b* positively regulated ABA-mediated drought resistance [48]. In this study, our phenotypic assay of the transgenic lines, *pyr1pyl124* quadruple mutant, and *Col-0* plant showed that *TaHsfC3-4* was a positive regulator in pleiotropic ABA signaling including cotyledon greening (Figure 4) and root growth (Figure 5). Furthermore, overexpression of *TaHsfC3-4* conferred drought tolerance in transgenic plants (Figure 6). In brief, TaHsfC3-4 enhances tolerance to drought through the ABA signaling pathway.

### 3.3. TaHsfC3-4 and TaHsfA2-11 Might Work Cooperatively to Regulate Drought-Responsive Gene Expression

Evidence suggests that drought-responsive TFs interact with each other as well as components of other stress pathways to regulate target genes’ expression [4]. Basic leucine zipper protein ABF2 have been shown to physically interact with DREB2C, an AP2-domain protein [49]. It was also reported that there were similar interactions between AREB/ABFs and NACs. For instance, ANAC096 interacts with ABF2/AREB1 and ABF4/AREB2 under dehydration and osmotic stress [50]. In the current study, we showed that TaHsfA2-11 was identified as one of the potential TaHsfC3-4-interacting proteins and enhanced drought tolerance in transgenic plants (Figure 7). Previous studies have shown that HsfA2 is an important regulatory factor during heat stress and forms heterodimers with HsfA1 resulting in the synergistic transcriptional activation of heat stress response genes [51]. However, there is growing evidence that the HsfA2 subclass is also involved in various other abiotic stresses, including drought. More recently, *TaHsfA2e-5D* was reported to be highly induced by drought, cold, and salinity. Overexpression of *TaHsfA2e-5D* in *Arabidopsis* improved both thermotolerance and drought tolerance [52]. Taken together, our data presented here not only demonstrate the roles of Hsf in drought, especially the members of the HsfC3 subclass, but also provides a target locus for marker-assisted selection breeding to improve drought tolerance in wheat.

## 4. Materials and Methods

### 4.1. Plant Materials and Growth Conditions

Cang6005, a semi-winter wheat cultivar, was used for gene cloning and expression analysis and was cultivated in Hoagland nutrient solution or soil mix. In addition, Columbia-0 (*Arabidopsis thaliana*) and *pyr1pyl1pyl2pyl4* (*pyr1pyl124*) quadruple mutant (provided by Prof. Hongtao Liu, National Key Laboratory of Plant Molecular Genetics, CAS Center for Excellence in Molecular Plant Sciences, Institute of Plant Physiology and Ecology, Chinese Academy of Sciences), used for genetic transformation and phenotypic analysis, were seeded in the same pot containing nutrient soil. *Nicotiana benthamiana*, used for observation of the fluorescence signal, was cultivated in soil mix. All the plants materials mentioned above were grown in a controlled environment under day/night conditions of 16 h/8 h (70–100 μmol·m^−2^·S^−1^), 25 °C, and 50–60% relative humidity. 

### 4.2. Bioinformatics Analysis of TaHsfC3-4

The protein sequences in different species used for multiple sequence alignment analysis were downloaded from the NCBI website (https://www.ncbi.nlm.nih.gov/). Evolutionary analyses were conducted using MEGA X (https://www.megasoftware.net, accessed on 18 December 2023). Multiple sequence alignment analysis was carried out by the DNAMAN 8.0 software. Protein structure analysis was performed using the online tool of SWISS-MODEL (https://swissmodel.expasy.org/, accessed on 3 January 2024). *Cis*-acting elements within approximately 2855 bp of the upstream promoter region of *TaHsfC3-4* were predicted by using PlantCARE database [34].

### 4.3. Isolation of Total RNA and Identification of TaHsfC3-4

For tissue-specific expression patterns of *TaHsfC3-4*, tissues of wheat at different growth and development stages were collected for RNA sample isolation. In order to examine the expression levels of *TaHsfC3-4* during the seed maturation stage, the total RNA was extracted from the tissue of wheat at different days after anthesis. In addition, to determine whether the expression of *TaHsfC3-4* was induced by 200 µM ABA and 20% PEG6000 treatments in wheat, the roots and the second-expanded leaves of young wheat seedlings were harvested separately after various times of treatment for the extraction of RNA. All the above RNA were extracted by using the TRIpure (Aidlab, Beijing, China), according to the manufacturer’s protocol, and its concentration was measured using NanoDrop2000 (Thermo Fisher Scientific, Waltham, MA, USA). Then, 1 µg of purified RNA was used as template to reverse-transcribe to cDNA. In addition, to detect *TaHsfC3-4* in transgenic plants, total RNA was extracted from three-week-old transgenic *Arabidopsis* plants, and semi-quantitative RT-PCR was performed with specific primers designed by using Primer 5.0 software. The specific primers are listed in Appendix A.

### 4.4. Gene Expression Analysis Using RT-qPCR

RT-qPCR was used to explore the tissue expression pattern and the induction of the *TaHsfC3-4* gene during ABA and PEG treatments. The experiment was performed with SYBR Green Real-time PCR Master Mix (TaKaRa, Dalian, China) according to the manufacturer’s instructions (Bio-Rad, Hercules, CA, USA). Referring to our previous research methods [53], *TaRP15* was selected as internal control to normalize the data. The relative expressions of genes were calculated using the average values of 2^−ΔΔct^. The analysis of each sample was repeated three times, and the results are presented as means and standard deviations. SPSS 19.0 software was used to analyze the statistical significance. The primers used are listed in Appendix A. 

### 4.5. Subcellular Localization

For subcellular localization assays, the *TaHsfC3-4* coding sequence without a stop codon was amplified and inserted into the pJIT1-hGFP expression vector driven by the *CaMV35S* promoter. This construct and empty plasmid were transformed into *Agrobacterium tumefaciens* GV3101 cells and infiltrated into *N. benthamiana* leaves. Then, the plants were grown for another two days. Before imaging, the nucleus of *N. benthamiana* leaves was stained with 1 mg·mL^−1^ DAPI dye (4′,6-diamidino-2-phenylindole). Fluorescence signals in the transformed *N. benthamiana* leaves were observed with a confocal laser scanning microscope (META510, Zeiss, Oberkochen, Germany).

### 4.6. Transactivation Activation Analysis in Yeast

For transcriptional activation activity assays, the full-length CDS of *TaHsfC3-4* was constructed into the yeast pGBKT7 vector and then introduced into *Saccharomyces cerevisiae* strain AH109 following the yeast protocol handbook (TakaRa, Dalian, China). The transformants were spotted onto the yeast synthetic drop-out medium (SD/-Trp and SD/-Trp/-His/-Ade) and incubated at 30 °C for three days before observation. The transcriptional activation activities were evaluated according to yeast growth status and the activity of α-galactosidase.

### 4.7. Generation of Transgenic Plants

To generate the complemented lines of *TaHsfC3-4* in *pyr1pyl124* quadruple mutant and transgenic plants overexpressing *TaHsfA2-11* in *Col-0*, the coding regions of *TaHsfC3-4* and *TaHsfA2-11* were cloned by PCR and inserted into the pCAMBIA1300 vector. The recombinant plasmids were introduced into *Agrobacterium tumefaciens* GV3101 cells. According to the classical floral dip method, genetic transformation was performed at the stage of flower-budding in *pyr1pyl124* quadruple mutant and *Col-0*. The transgenic seeds were selected on MS-agar plates containing 30 µg·mL^−1^ hygromycin and detected by PCR. Homozygous T3 transgenic lines were selected for phenotypic observation.

### 4.8. Phenotypic Assay under ABA Treatment

For cotyledon greening assays, sterilized seeds were grown on 1/2 MS-agar plates with or without 0.5 μM ABA and kept at 4 °C for 3 days. Green cotyledons were recorded on the eighth day after the plates were transferred to a growth chamber at 25 °C under a 16 h light/8 h dark photoperiod with a light intensity of ~100 μE m^−2^ s^−1^. For root length assays, four-day-old seedlings were transferred onto 1/2 MS-agar plates supplemented or not with 10 μM ABA and grown vertically for another four days. Subsequently, the root elongation was measured, and photographs of seedlings were taken at the specified time points.

### 4.9. Drought Tolerance Assay 

Sterilized seeds of various genotypes were first stratified for 3 days at 4 °C in the dark and then germinated for 7 days on 1/2 MS-agar plates with 1% sucrose. Seven-day-old seedlings were transplanted to the soil mix and grew for fourteen days with normal watering. Subsequently, the plants were subjected to drought stress by withholding watering for 14 days. After rewatering for 3 days, photographs were taken and the rosette leaves of different lines were collected to measure the physiological indexes. 

### 4.10. Measurements of Physiological Indexes

Chlorophyll contents in the leaves of seedlings were measured photometrically, and the method was slightly improved [54]. In brief, 1 g of leaves was put into a 100 mL test tube containing 20 mL mixture (acetone: pure methanol: ddH_2_O, 4.5:4.5:1.0) and homogenized. The homogenate was analyzed immediately after filtration. Then, OD_645_ and OD_663_ of the filtrate were spectrophotometrically determined for chlorophyll a (C_a_) and chlorophyll b (C_b_), respectively. Finally, the content of total chlorophyll (C_a+b_) was obtained by the following formula: C_a+b_ = 8.02 × OD_663_ + 20.20 × OD_645_.

For determination of SOD activity, put 1 g of leaves into a mortar and add 5 mL phosphoric acid buffer (0.1 mM EDTA, 0.3% Triton X-100, and 4% polyvinylpolyrrolidone). Grind the samples thoroughly. Filter the homogenate and centrifuge for 20 min with 10,500× *g*. The supernatant was the crude enzyme liquid. Take 1 mL supernatant into test tube containing 3 mL reaction mixture (13.05 mM DL-Methionine, 0.1 μM EDTA, 75 μM nitro-blue tetrazolium, 2 μM riboflavin). Then, put the test tube in a light incubator and shine it for 10 min. The test tube without crude enzyme solution was used as the control. OD_500_ value of the control (A_ck_) and the sample mixture (A_s_) were quickly determined. Measure the protein content (pro) of the supernatant. The activity of SOD was calculated according to the following formula: U/mg·pro = (A_ck_ − A_s_)/(0.5 × A_ck_ × pro).

The activity of POD was measured by guaiacol method. Put 1 g of leaves into a mortar and add 5 mL potassium dihydrogen phosphate buffer (20 mM, pH 6.0) with a little quartz sand to grind the samples. Transfer the mixture to a test tube and centrifuge for 10 min at 10,000× *g*. The supernatant was the crude enzyme liquid. Then the change (A_1_ − A_0_) of OD_470_ value was measured. Measure the protein content (pro) of the supernatant. The activity of POD was calculated by the following formula: U/mg·pro·min = (A_1_ − A_0_) × V_t_/pro·V_s_·0.01. V_t_ is the total volume of crude enzyme liquid and V_s_ is the volume of enzyme liquid taken at the experiment. 

Thiobarbituric acid (TBA) reactive substances assay was used to determine the content of MDA [47]. Briefly, put 1 g of leaves into a mortar and add 10 mL trichloroacetic acid (10%) with a little quartz sand to grind the samples. Transfer the mixture to the test tube and centrifuge for 10 min at 4000× *g*. Take 2 mL the supernatant into a new test tube and add 2 mL 0.6% thiobarbituric acid prepared with 10% trichloroacetic acid. Then, boil the mixture in water for 15 min. Cool quickly and centrifuge for 10 min at 4000× *g*. Measure OD_532_, OD_600_, and OD_450_ values of the supernatant. Measure the protein content (pro) of the supernatant. The content of MDA was obtained following the formula: nmol/mg·pro = 6.45 × (OD_532_ − OD_600_) − 0.56 × OD_450_/pro. 

### 4.11. Yeast Two-Hybrid Assay 

To study protein interactions in yeast, the coding sequences of *TaHsfC3-4* and *TaHsfA2-11* were cloned into the pGBKT7 and pGADT7 vectors, respectively. The bait and prey constructs were co-transformed into *Saccharomyces cerevisiae* strain AH109 by the lithium acetate method according to the manufacturer’s guidelines. Successfully transformed colonies were identified on yeast SD medium lacking Leu and Trp. Then, the transformants were spotted onto the yeast synthetic drop-out medium (SD/-Trp/-Leu and SD/-Trp/-Leu/-His/-Ade) and incubated at 30 °C for three days before observation. The possible interaction was evaluated according to yeast growth status and the activity of α-galactosidase.

### 4.12. LCI Assay

The LCI assays for the interaction between TaHsfC3-4 and TaHsfA2-11 were performed in *N. benthamiana* leaves. The coding sequences of *TaHsfC3-4* and *TaHsfA2-11* were, respectively, fused with the N-terminal and C-terminal parts of the luciferase reporter gene to obtain the TaHsfC3-4-nLUC and cLUC-TaHsfA2-11 constructs. Constructs including empty vectors were transformed into *Agrobacterium tumefaciens* GV3101 cells and infiltrated into *N. benthamiana* leaves. Then, the plants were grown for another two days, and the infiltrated leaves were analyzed for LUC activity by using chemiluminescence imaging (Tanon 5200).

## Figures and Tables

**Figure 1 ijms-25-00977-f001:**
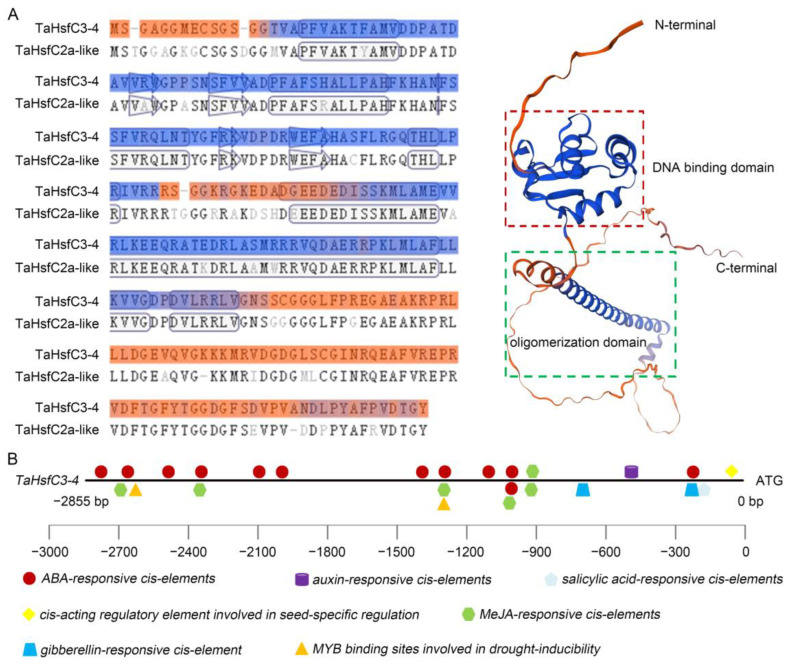
Structural analysis of TaHsfC3-4 protein and predicted *cis*-elements on the *TaHsfC3-4* promoter. (**A**) Schematic diagram of the TaHsfC3-4 protein. The protein sequence of TaHsfC2a-like (UniProtKB/TrEMBL, A0A3B6J0E8) was used as a template for alignment and structure prediction of TaHsfC3-4 with the online tool of SWISS-MODEL (https://swissmodel.expasy.org/, accessed on 3 January 2024). (**B**) Predicted *cis*-elements on the *TaHsfC3-4* promoter. The bold black line and the colorful icons represents the promoter and different *cis*-elements, respectively. Promoter sequences upstream of the translational start site are marked by negative numbers.

**Figure 2 ijms-25-00977-f002:**
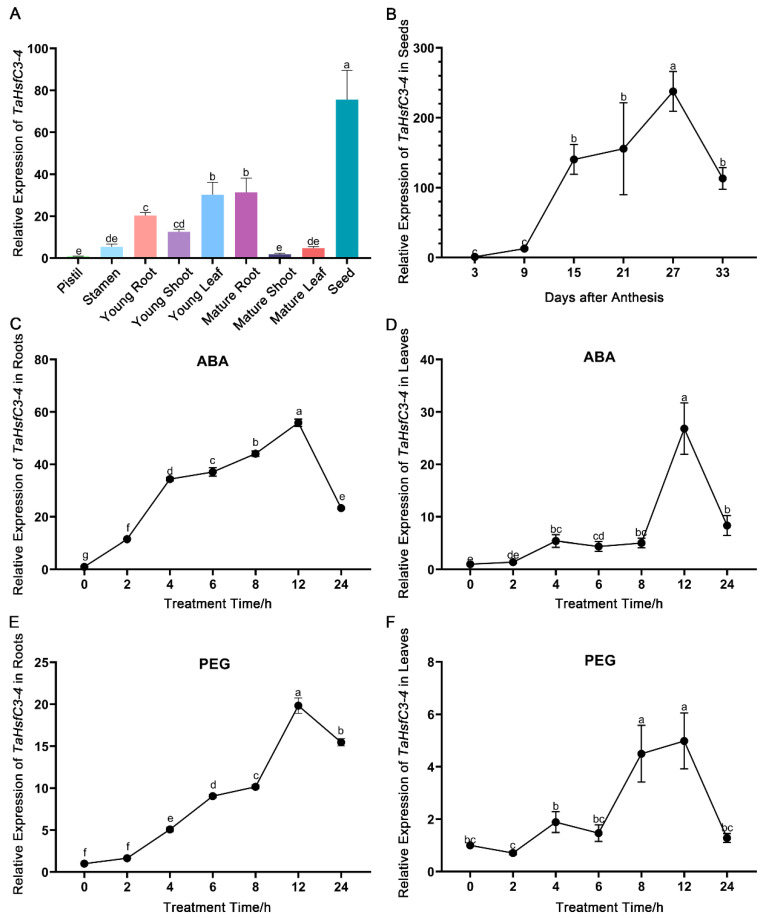
Expression characteristics analysis of *TaHsfC3-4*. (**A**) Tissue-specific expression patterns of *TaHsfC3-4*. RNA samples derived from nine tissues of wheat at different growth and development stages. (**B**) Expression levels of *TaHsfC3-4* during seed maturation stage. Total RNA was extracted from tissue of wheat during seed maturation stage and analyzed by RT-qPCR. (**C**,**D**) Time-course expression of *TaHsfC3-4* in roots (**C**) and leaves (**D**) treated with or without 200 μM ABA. (E and F) Time-course expression of *TaHsfC3-4* in roots (**E**) and leaves (**F**) in the presence or absence of 20% PEG6000. The roots and shoots of wheat young seedlings were harvested separately after various times of treatment. The control without ABA or PEG6000 was established in parallel. The wheat *TaRP15* gene was used to normalize relative expression levels. All data represent means ± standard deviation (SD) of three biological replicates. Different lowercase letters above the bars as determined by one-way ANOVA indicate significant differences at the *p* < 0.05 level.

**Figure 3 ijms-25-00977-f003:**
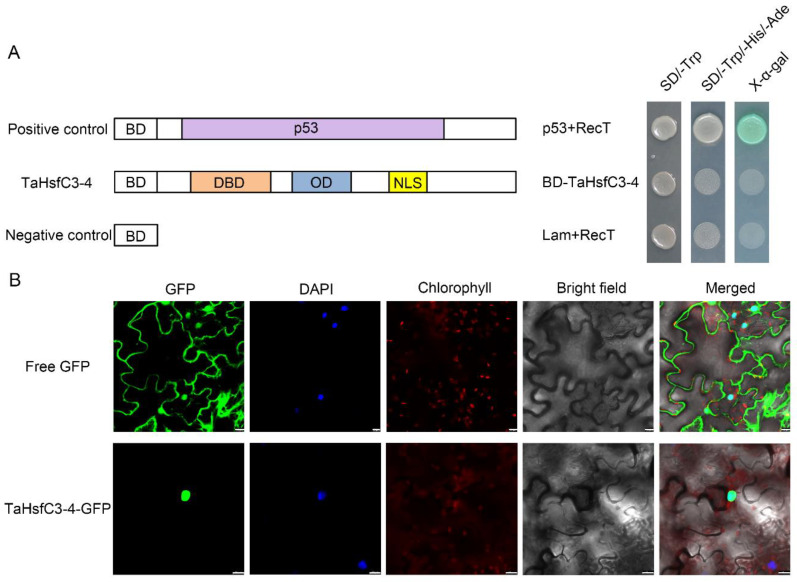
Transcriptional activity analyses and subcellular localization of TaHsfC3-4. (**A**) Transcriptional activation analysis of TaHsfC3-4 in yeast. TaHsfC3-4 proteins were fused to the GAL4 DNA-binding domain (BD) and then transformed into *Saccharomyces cerevisiae* strain AH109. The transformants were spotted onto the yeast synthetic drop-out medium (SD/-Trp and SD/-Trp/-His/-Ade) and incubated at 30 °C for three days. p53 and RecT, positive control; lamin and RecT, negative control. (**B**) Subcellular localization of TaHsfC3-4 in the leaf epidermal cells of *N. benthamiana*. The 35S::GFP (control) and 35S::TaHsfC3-4-GFP were transformed into the leaf epidermal cells of *N. benthamiana* and visualized by DAPI staining. Bars = 20 μm.

**Figure 4 ijms-25-00977-f004:**
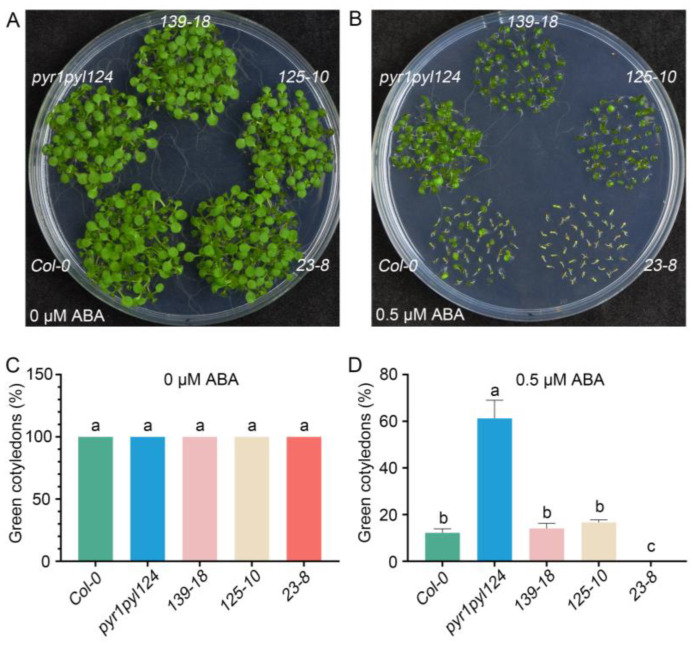
Overexpression of *TaHsfC3-4* complemented the ABA-hyposensitive phenotypes of the *pyr1pyl124* quadruple mutant in cotyledon greening. (**A**–**D**) The cotyledon greening assay of *TaHsfC3-4* transgenic lines, *pyr1pyl124* quadruple mutant, and *Col-0* on normal medium (**A**,**C**) or medium supplemented with 0.5 μM ABA (**B**,**D**). Values in (**C**,**D**) represent means ± SD obtained from three replicates with 50 seeds per replicate. Different lowercase letters above the bars as determined by one-way ANOVA indicate significant differences at the *p* < 0.05 level.

**Figure 5 ijms-25-00977-f005:**
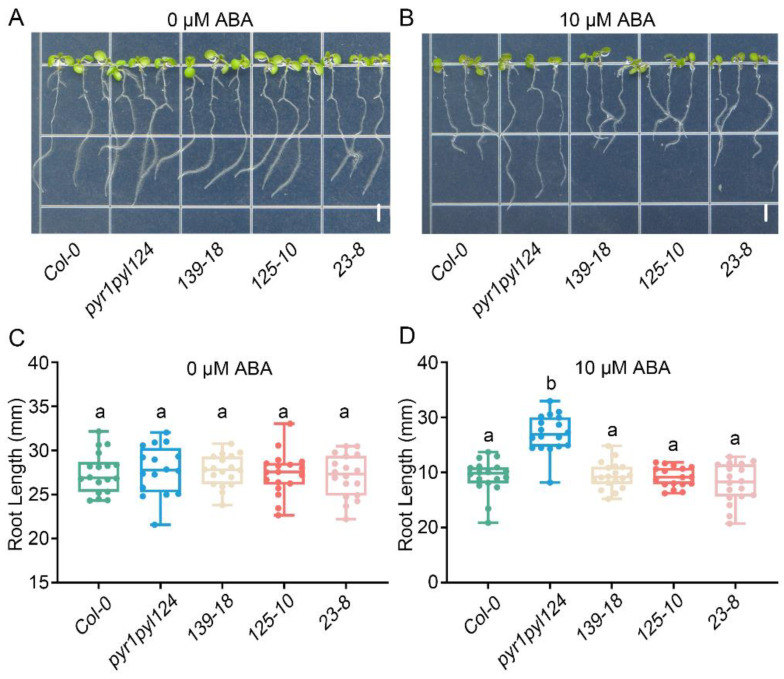
Overexpression of *TaHsfC3-4* complemented the ABA-hyposensitive phenotypes of the *pyr1pyl124* quadruple mutant in root elongation. (**A**–**D**) Root elongation assay of *TaHsfC3-4* transgenic lines, *pyr1pyl124* quadruple mutant, and *Col-0* on normal medium (**A**,**C**) or medium supplemented with 10 μM ABA (**B**,**D**). Values in (**C**,**D**) represent means ± SD obtained from three replicates with 15 plants per replicate. Different lowercase letters above the bars as determined by one-way ANOVA indicate significant differences at the *p* < 0.05 level. Bars = 4 mm.

**Figure 6 ijms-25-00977-f006:**
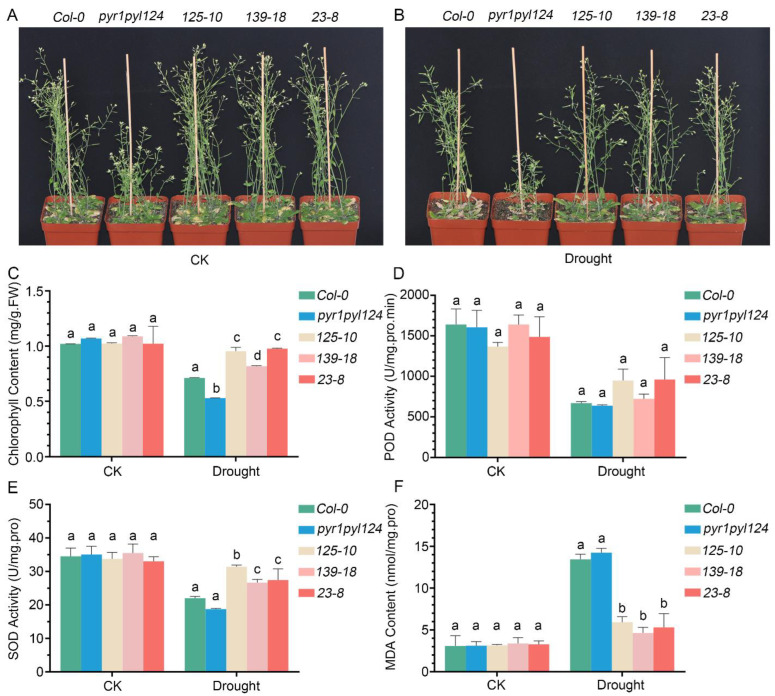
TaHsfC3-4 enhanced drought-stress resistance of the *pyr1pyl124* quadruple mutant. (**A**,**B**) Drought-tolerant phenotypes of soil-grown *TaHsfC3-4* transgenic lines, *pyr1pyl124* quadruple mutant, and *Col-0*. Three-week-old plants were dehydrated for 14 days and then rewatered for 3 days before phenotyping. (**C**) Chlorophyll content of *TaHsfC3-4* transgenic lines, *pyr1pyl124* quadruple mutant, and *Col-0* under drought treatment or not. (**D**,**F**) POD, SOD activities, and MDA content of *TaHsfC3-4* transgenic lines, *pyr1pyl124* quadruple mutant, and *Col-0* under drought treatment or not. Values in (**C**–**F**) represent means ± SD obtained from three replicates with 15 plants per replicate. Different lowercase letters above the bars as determined by one-way ANOVA indicate significant differences at the *p* < 0.05 level.

**Figure 7 ijms-25-00977-f007:**
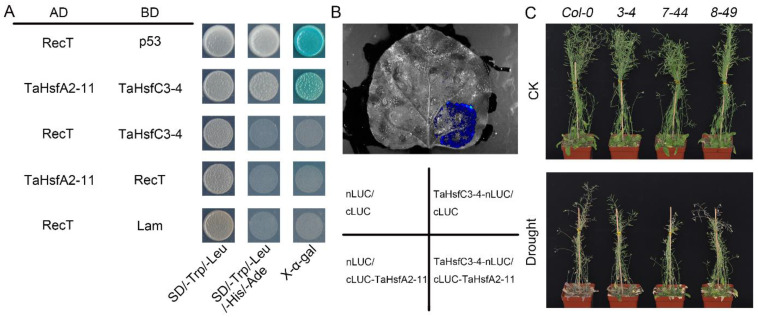
TaHsfA2-11 interacts with TaHsfC3-4 and improves drought tolerance of transgenic *Arabidopsis*. (**A**) Y2H assays showing the interaction relationships between TaHsfC3-4 and TaHsfA2-11. The TaHsfC3-4 protein is fused to the GAL4 DNA-binding domain (BD), and the TaHsfA2-11 proteins are fused to the GAL4 activation domain (AD). The transformants were grown on the control medium DDO (SD/–Trp/–Leu) and selective medium QDO (SD/–Trp/–Leu/–His/–Ade). (**B**) LCI assays of TaHsfC3-4-nLUC and cLUC-TaHsfA2-11 in leaves of *N. benthamiana*. TaHsfC3-4-nLUC and cLUC-TaHsfA2-11 were transiently expressed in the leaf epidermal cells of *N. benthamiana* for two days. Luminescence was observed by using a cooled CCD imaging apparatus. (**C**) Drought-tolerant phenotypes of soil-grown *TaHsfA2-11* overexpressing transgenic *Arabidopsis* lines and *Col-0*. Three-week-old plants were dehydrated for 14 days and then rewatered for 3 days before phenotyping.

## Data Availability

Data are contained within the article and Appendix A.

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
