# Peer review of "Upregulation of Wheat Heat Shock Transcription Factor *TaHsfC3-4* by ABA Contributes to Drought Tolerance"

_ijms, 2024, doi:10.3390/ijms25020977_

Round 1

Reviewer 1 Report

Comments and Suggestions for Authors

Comment 01:

In the manuscript titled "Up-Regulation of Wheat Heat Shock Transcription Factor TaHsfC3-4 by ABA Contributes to Drought Tolerance," submitted to the International Journal of Molecular Sciences (Manuscript ID: ijms-2774464), the authors explore a significant facet of agricultural biotechnology. This research delves into the molecular mechanisms by which wheat (Triticum aestivum), a vital global crop, combats drought stress, specifically through the lens of heat shock transcription factors (Hsfs). The study's focus on the subclass HsfC member, TaHsfC3-4, is particularly noteworthy due to the previously limited understanding of its biological functions in stress response.

The manuscript provides compelling evidence of TaHsfC3-4's involvement in abscisic acid (ABA)-dependent pathways of drought resistance, a finding that enriches our understanding of plant stress physiology. The authors' methodical approach, utilizing transgenic Arabidopsis models and yeast two-hybrid screening, substantiates the role of TaHsfC3-4 in enhancing drought tolerance. This discovery is not only a significant advancement in the field of plant molecular biology but also holds substantial socio-economic implications. Enhancing drought resistance in wheat could lead to more resilient agricultural practices, directly benefiting global food security in the face of climate change. Furthermore, the potential for marker-assisted selection breeding highlighted in this study opens avenues for crop improvement strategies that are both efficient and ecologically sustainable. The research, thus, stands as a pivotal contribution to both scientific understanding and practical agricultural applications.

Comment 02:

Overall, the experimental methodologies applied in this study are not only appropriate but also executed with a high degree of scientific diligence, greatly strengthening the validity and impact of the findings. This level of experimental excellence sets a benchmark for future studies in the field and contributes significantly to the robustness and reproducibility of the results. The meticulous approach to experimental design and execution in this study significantly enhances its contribution to the field of molecular plant biology and agricultural biotechnology.

Comment 03:

A point that warrants further clarification pertains to the experimental use of two distinct concentrations of abscisic acid (ABA), namely 0.5 µM and 10 µM. The rationale behind the selection of these specific concentrations and the implications of such choices for the experimental outcomes need to be explicitly addressed.

It is crucial to understand whether these concentrations were chosen based on previous empirical findings, theoretical predictions, or preliminary experiments.

Comment 04:

The "Materials and Methods" section requires significant elaboration for clarity and scientific rigor. For example, the current brevity, as exemplified by the cursory description of chlorophyll content determination ("measured photometrically"), undermines the reproducibility and validation of the research.

A detailed "Materials and Methods" section should include comprehensive information on experimental design, materials used, and step-by-step procedures. This includes specifying types of samples, concentrations of reagents, equipment models, settings, and the protocols for each assay or measurement. For instance, in measuring chlorophyll content, it's crucial to detail the type of chlorophyll measured (a, b, or total), the extraction solvent, photometric method, wavelength of absorbance, and the equations used for calculating concentrations.

Such details are fundamental in scientific research as they allow peers to replicate the study, validate findings, and compare results across different studies. The current lack of detail hinders these critical aspects of scientific communication.

While chlorophyll content measurement is just one example, a similar level of detail is needed across all aspects of the "Materials and Methods" section to ensure the study's reliability and reproducibility.

Comment 05:

A more comprehensive phylogenetic tree within the article, incorporating a wider range of HsfC sequences from diverse species, is recommended. This inclusion would provide a deeper understanding of the evolutionary history and functional diversification of the HsfC family. It should detail the methodologies for phylogenetic analysis, including sequence selection criteria, alignment tools, and tree construction models and algorithms. Such an expanded phylogenetic analysis would offer valuable insights into the role of HsfCs in plant stress responses, enhancing the study's contribution to molecular evolution and plant biology.

Comment 06:

The choice of the actin2 gene as the reference gene in your study raises curiosity, particularly given the availability of various alternative reference genes in Arabidopsis thaliana. It would be beneficial to understand the specific rationale behind this selection. While this does not question the constitutive nature of the actin2 gene in your study, elucidating the decision-making process could offer valuable insights into your experimental design, particularly in terms of gene expression normalization. Understanding the basis for selecting actin2 over other potential candidates could enhance the scientific community's appreciation of the methodological choices made in your research.

Comment 07:

The manuscript currently cites only two references from 2023. Expanding the reference list to include more recent literature would significantly enhance the context and relevance of your work.

Comment 07:

Suggestions for corrections:

Please change "colling" to "cooling".

Please change "have been functional analyzed" to "have been functionally analyzed".

Please change "expresssion" to "expression".

Please change "coinfiltrated" to "co-infiltrated".

Please change "Realtime" to "Real-time".

Please change "denateration" to "denaturation".

Please change "pyr1pyl124 quadruple mutant show" to "pyr1pyl124 quadruple mutant shows".

Comment 08:

Your work is indeed interesting, and it is important to note that my critiques are not intended to diminish its quality. They are aimed at enhancing the clarity and depth of the study, which in turn can help to solidify its contributions to the field. The suggestions provided are meant to strengthen the manuscript and ensure that its scientific merit is clearly communicated and appreciated by the wider research community.

Comments on the Quality of English Language

Minor editing of English language required

Author Response

Responds to the reviewer1 comments:

Comment 01:

In the manuscript titled "Up-Regulation of Wheat Heat Shock Transcription Factor TaHsfC3-4 by ABA Contributes to Drought Tolerance," submitted to the International Journal of Molecular Sciences (Manuscript ID: ijms-2774464), the authors explore a significant facet of agricultural biotechnology. This research delves into the molecular mechanisms by which wheat (Triticum aestivum), a vital global crop, combats drought stress, specifically through the lens of heat shock transcription factors (Hsfs). The study's focus on the subclass HsfC member, TaHsfC3-4, is particularly noteworthy due to the previously limited understanding of its biological functions in stress response.

The manuscript provides compelling evidence of TaHsfC3-4's involvement in abscisic acid (ABA)-dependent pathways of drought resistance, a finding that enriches our understanding of plant stress physiology. The authors' methodical approach, utilizing transgenic Arabidopsis models and yeast two-hybrid screening, substantiates the role of TaHsfC3-4 in enhancing drought tolerance. This discovery is not only a significant advancement in the field of plant molecular biology but also holds substantial socio-economic implications. Enhancing drought resistance in wheat could lead to more resilient agricultural practices, directly benefiting global food security in the face of climate change. Furthermore, the potential for marker-assisted selection breeding highlighted in this study opens avenues for crop improvement strategies that are both efficient and ecologically sustainable. The research, thus, stands as a pivotal contribution to both scientific understanding and practical agricultural applications.

Responds: Thank you for your support and understanding of this paper, which will be of great help to our future research.

Comment 02:

Overall, the experimental methodologies applied in this study are not only appropriate but also executed with a high degree of scientific diligence, greatly strengthening the validity and impact of the findings. This level of experimental excellence sets a benchmark for future studies in the field and contributes significantly to the robustness and reproducibility of the results. The meticulous approach to experimental design and execution in this study significantly enhances its contribution to the field of molecular plant biology and agricultural biotechnology.

Responds: Thank you for your approval of this article.

Comment 03:

A point that warrants further clarification pertains to the experimental use of two distinct concentrations of abscisic acid (ABA), namely 0.5 µM and 10 µM. The rationale behind the selection of these specific concentrations and the implications of such choices for the experimental outcomes need to be explicitly addressed.

It is crucial to understand whether these concentrations were chosen based on previous empirical findings, theoretical predictions, or preliminary experiments.

Responds: In fact, we refer to previous articles, where most researchers observed seed germinationa and cotyledon greening under 0.5 µM ABA. However, root elongation is usually observed under 10 µM ABA. The rationale behind the selection of these specific concentrations requires us to spend more time and experience to find a lot of literature or consult senior experts. All in all, thank you for your valuable advice, prompting us to learn more about ABA.

Comment 04:

The "Materials and Methods" section requires significant elaboration for clarity and scientific rigor. For example, the current brevity, as exemplified by the cursory description of chlorophyll content determination ("measured photometrically"), undermines the reproducibility and validation of the research.

A detailed "Materials and Methods" section should include comprehensive information on experimental design, materials used, and step-by-step procedures. This includes specifying types of samples, concentrations of reagents, equipment models, settings, and the protocols for each assay or measurement. For instance, in measuring chlorophyll content, it's crucial to detail the type of chlorophyll measured (a, b, or total), the extraction solvent, photometric method, wavelength of absorbance, and the equations used for calculating concentrations.

Such details are fundamental in scientific research as they allow peers to replicate the study, validate findings, and compare results across different studies. The current lack of detail hinders these critical aspects of scientific communication.

While chlorophyll content measurement is just one example, a similar level of detail is needed across all aspects of the "Materials and Methods" section to ensure the study's reliability and reproducibility.

Responds: We've expanded this section to include more detail.

Comment 05:

A more comprehensive phylogenetic tree within the article, incorporating a wider range of HsfC sequences from diverse species, is recommended. This inclusion would provide a deeper understanding of the evolutionary history and functional diversification of the HsfC family. It should detail the methodologies for phylogenetic analysis, including sequence selection criteria, alignment tools, and tree construction models and algorithms. Such an expanded phylogenetic analysis would offer valuable insights into the role of HsfCs in plant stress responses, enhancing the study's contribution to molecular evolution and plant biology.

Responds: In accordance with your request, we have supplemented the phylogenetic tree of HsfC in Figure S2.

Comment 06:

The choice of the actin2 gene as the reference gene in your study raises curiosity, particularly given the availability of various alternative reference genes in Arabidopsis thaliana. It would be beneficial to understand the specific rationale behind this selection. While this does not question the constitutive nature of the actin2 gene in your study, elucidating the decision-making process could offer valuable insights into your experimental design, particularly in terms of gene expression normalization. Understanding the basis for selecting actin2 over other potential candidates could enhance the scientific community's appreciation of the methodological choices made in your research.

Responds: Our selection of actin2 as a reference gene is based on the previous article “Characterization of the wheat heat shock factor TaHsfA2e-5D conferring heat and drought tolerance in Arabidopsis”.

Comment 07:

The manuscript currently cites only two references from 2023. Expanding the reference list to include more recent literature would significantly enhance the context and relevance of your work.

Responds: In this paper, we add more recent literature, for example “Wheat adaptation to environmental stresses under climate change: Molecular basis and genetic improvement. Mol Plant. 2023, 16: 1564-1589”.

Comment 07:

Suggestions for corrections:

Please change "colling" to "cooling".

Please change "have been functional analyzed" to "have been functionally analyzed".

Please change "expresssion" to "expression".

Please change "coinfiltrated" to "co-infiltrated".

Please change "Realtime" to "Real-time".

Please change "denateration" to "denaturation".

Please change "pyr1pyl124 quadruple mutant show" to "pyr1pyl124 quadruple mutant shows".

Responds: The corresponding content of the article has been revised.

Comment 08:

Your work is indeed interesting, and it is important to note that my critiques are not intended to diminish its quality. They are aimed at enhancing the clarity and depth of the study, which in turn can help to solidify its contributions to the field. The suggestions provided are meant to strengthen the manuscript and ensure that its scientific merit is clearly communicated and appreciated by the wider research community.

Responds: Thank you again for your appreciation of this article

Reviewer 2 Report

Comments and Suggestions for Authors

Here are my comments

- The methodology section needs expansion and clarification, especially regarding gene expression analysis and drought tolerance assays, to ensure reproducibility and clarity.

- Enhance the statistical analysis to provide more comprehensive data including effect sizes, confidence intervals, and a more detailed interpretation of the statistical significance.

- Broaden the study's scope to include ABA-independent pathways in the drought stress response for a more holistic understanding of the plant's stress mechanisms.

- A more detailed functional characterization of TaHsfC3-4 is required, exploring its molecular mechanism and physiological role in drought tolerance thoroughly.

- Investigate the interaction between TaHsfC3-4 and TaHsfA2-11 more deeply with experimental evidence to elucidate the role of this interaction in enhancing drought tolerance.

- Incorporate comparative analysis with other members of the HsfC family, both in wheat and other species, to distinguish the unique functions of TaHsfC3-4.

- Resolve inconsistencies in the presented data and ensure all results include adequate controls and sufficient replication for robustness.

- Reorganize the manuscript for clearer separation and logical flow between sections, especially distinguishing results from discussion.

- Expand the discussion to place the study within a broader scientific context and explore the implications of the findings for practical applications in agriculture.

- Simplify technical language and reduce jargon to make the manuscript more accessible to a wider audience, including non-specialists in the field.

Author Response

- The methodology section needs expansion and clarification, especially regarding gene expression analysis and drought tolerance assays, to ensure reproducibility and clarity.

Responds: In fact, we have expanded and rewritten the methods section to ensure the repeatability of the data, hoping to meet your requirements for this article.

- Enhance the statistical analysis to provide more comprehensive data including effect sizes, confidence intervals, and a more detailed interpretation of the statistical significance.

Responds: We have made corresponding changes in the article.

- Broaden the study's scope to include ABA-independent pathways in the drought stress response for a more holistic understanding of the plant's stress mechanisms.

Responds: We have made corresponding supplements in the paper, such as the case of Arabidopsis HsfA1b.

- A more detailed functional characterization of TaHsfC3-4 is required, exploring its molecular mechanism and physiological role in drought tolerance thoroughly.

Responds: In fact, we are considering using Chip-seq to understand downstream signaling networks regulated by TaHsfC3-4, but we have not yet obtained relevant overexpression lines, so unfortunately we cannot provide valid experimental evidence.

- Investigate the interaction between TaHsfC3-4 and TaHsfA2-11 more deeply with experimental evidence to elucidate the role of this interaction in enhancing drought tolerance.

Responds: We seriously considered the implications of interaction and started to do some genetic evidence, but it is still taking a long time to get experimental evidence, and we are sorry about that.

- Incorporate comparative analysis with other members of the HsfC family, both in wheat and other species, to distinguish the unique functions of TaHsfC3-4.

Responds: In accordance with your request, we have supplemented the phylogenetic tree of HsfC in Figure S2.

- Resolve inconsistencies in the presented data and ensure all results include adequate controls and sufficient replication for robustness.

Responds: We rechecked the results section and made sure the text description matched the figure presentation

- Reorganize the manuscript for clearer separation and logical flow between sections, especially distinguishing results from discussion.

Responds: Thanks for your suggestions, we have revised the full text to make it more logical and complete.

- Expand the discussion to place the study within a broader scientific context and explore the implications of the findings for practical applications in agriculture.

 Responds: In the rewritten section, we emphasized the particularity of TaHsfC3-4 in crops, especially wheat, in order to highlight its breeding value.

- Simplify technical language and reduce jargon to make the manuscript more accessible to a wider audience, including non-specialists in the field.

 Responds: Under your suggestion, we have sorted out the whole text and rewritten some sentences.

Reviewer 3 Report

Comments and Suggestions for Authors

The manuscript entitled "Up-Regulation of Wheat Heat Shock Transcription Factor TaHsfC3-4 by ABA Contributes to Drought Tolerance" aims to confirm the role of TaHsfC3-4 in response to drought stress.

The article is appropriately structured and includes seven figures. The methodology is set correctly. Fifty-four literary sources are included in the literature. TaHsfC3-4 encodes a protein containing 274 amino acids and has the basic characteristics of HsfC. Based on the previous omics data of the scientific group, they identified a class C encoding gene TaHsfC3-4 and analysed its biological function in transgenic plants. The experiments showed that TaHsfC3-4 was constitutively expressed in many wheat tissues and induced during seed maturation. The observed expression profile of TaHsfC3-4 suggests that this Hsf may be involved in the regulation pathway depending on the ABA of drought resistance. The authors found that TaHsfC3-4 was localised in the nucleus but lacked transcriptional activation activity. The main contribution of this study is that it confirms the role of TaHsfC3-4 in response to drought stress and provides a target locus for marker-assisted breeding to improve drought tolerance in wheat.

The overall research and results obtained by the authors are well presented.

Based on the above, the article "Up-Regulation of Wheat Heat Shock Transcription Factor TaHsfC3-4 by ABA Contributes to Drought Tolerance" with authors Zhenyu Ma, Baihui Zhao, Huaning Zhang, Shuonan Duan, Zihui Liu, Xiulin Guo, Xiangzhao Meng, Guoliang Li contains interesting scientific facts, representing news for the science. It can be published in the International Journal of Molecular Sciences. It fully corresponds to his scope.

Some remarks and questions:

Line 117-119: The sentence must be clarified!

The quality of Figure 1A is low!

Standardise RT-qPCR in all places!

Line 210: "...normal MS medium (Figure 5A and 5C)", but on the figure is written CK?!

Figures 6C and 6E: Please check the lowercase letters above the bars in drought conditions!

Author Response

Responds to the reviewer3 comments:

Some remarks and questions:

Line 117-119: The sentence must be clarified!

Responds: We have rewritten this sentence.

The quality of Figure 1A is low!

Responds: We have changed the Figure 1A in the article.

Standardise RT-qPCR in all places!

Responds: In accordance with your request, we have corrected all non-standard descriptions in the article.

Line 210: "...normal MS medium (Figure 5A and 5C)", but on the figure is written CK?!

Responds: In order to eliminate the doubts of you and readers, we unify the description in the article.

Figures 6C and 6E: Please check the lowercase letters above the bars in drought conditions!

Responds: We double-check and ensure that the significance analysis is true and reliable.

Thank you again for the comments and suggestions!